# Technological Features for Controlling Steel Part Quality Parameters by the Method of Electrospark Alloying Using Carburezer Containing Nitrogen—Carbon Components

**DOI:** 10.3390/ma15176085

**Published:** 2022-09-02

**Authors:** Oksana P. Gaponova, Viacheslav B. Tarelnyk, Bogdan Antoszewski, Norbert Radek, Nataliia V. Tarelnyk, Piotr Kurp, Oleksandr M. Myslyvchenko, Jacek Hoffman

**Affiliations:** 1Applied Material Science and Technology of Constructional Materials Department, Sumy State University, 40007 Sumy, Ukraine; 2Technical Service Department, Sumy National Agrarian University, H. Kondratiieva Str., 160, 40021 Sumy, Ukraine; 3Laser Research Centre, Faculty of Mechatronics and Mechanical Engineering, Kielce University of Technology, Al. Tysiąclecia P.P. 7, 25-314 Kielce, Poland; 4Physical Chemistry of Inorganic Materials Department, I. M. Frantsevich Institute for Problems in Materials Science, 04060 Kyiv, Ukraine; 5Department of Experimental Mechanics, Institute of Fundamental Technological Research, Polish Academy of Sciences, ul. Pawińskiego 5B, 02-016 Warsaw, Poland

**Keywords:** electrospark alloying, coatings, roughness, structure, microhardness, continuity, X-ray diffraction analysis, nitriding, nitrocarburizing, carburization

## Abstract

A new method of surface modification based on the method of electrospark alloying (ESA) using carburizer containing nitrogen—carbon components for producing coatings is considered. New processes have been proposed that include the step of applying saturating media in the form of paste-like nitrogenous and nitrogenous-carbon components, respectively, onto the surface without waiting for those media to dry, conducting the ESA process with the use of a steel electrode-tool, as well as with a graphite electrode-tool. Before applying the saturating media, an aluminium layer is applied onto the surface with the use of the ESA method at a discharge energy of W_p_ = 0.13–6.80 J. A saturating medium in the form of a paste was applied to the surfaces of specimens of steel C22 and steel C40. During nitriding, nitrocarburizing and carburization by ESA (CESA) processes, with an increase in the discharge energy (W_p_), the thickness, micro hardness and continuity of the “white layer” coatings, as well as the magnitude of the surface roughness, increase due to saturation of the steel surface with nitrogen and/or carbon, high cooling rates, formation of non-equilibrium structures, formation of special phases, etc. In the course of nitriding, nitrocarburizing and CESA processing of steels C22 and C40, preliminary processing with the use of the ESA method by aluminum increases the thickness, microhardness and continuity of the “white layer”, while the roughness changes insignificantly. Analysis of the phase composition indicates that the presence of the aluminum sublayer leads to the formation of the aluminum-containing phases, resulting in a significant increase in the hardness and, in addition, in an increase in the thickness and quality of the surface layers. The proposed methods can be used to strengthen the surface layers of the critical parts and their elements for compressor and pumping equipment.

## 1. Introduction

One of the most important issues in modern mechanical engineering is the improvement of technology for production of critical parts of a new generation of tools that operate under extreme environmental conditions and at constantly increasing operating parameters of pressure, temperature, speed, radiation exposure, etc.

In the course of a machine’s operation, all its parts, working bodies and their elements are subject to various types of wear, as a result of which serious accidents can occur. Degradation of a machine part usually starts from its surface. Therefore, technologists from all countries are paying more and more attention to the development of technologies which improve the quality of surface layers and the creation of composite materials which are strong and have protective properties.

Fundamentally new materials are needed, with increased wear resistance. This may begin with naturally occurring substances, from which follows the natural physical substance from the vapor phase (PVD) [1], the chemical substance from the vapor phase [2], the hybrid physico-chemical substance from the vapor phase [3] and thermal spraying methods [4]. Hardening technologies, such as chemical-thermal treatment (CTT) are critically important [5].

The main CTT methods are carburizing (carbon saturation) [6], nitriding (nitrogen saturation), carried out in recent years by the ion-plasma method in a glow discharge [7], which, unlike traditional gas nitriding in furnaces, reduces the process duration by 1.5–2 times and is more environmentally friendly [8] and nitrocarburizing (saturation with carbon and nitrogen) [9].

Often these CTT methods are carried out using expensive technogenic and environmentally hazardous equipment that may have a different design, environment (solid, liquid and gaseous), exposure temperature and processing time, often measured in tens of hours. According to [10], CTT methods are among the most common risk factors.

Thus, there is a need to develop a new, cheaper, environmentally friendly and technologically safe method of processing parts, an alternative to the CTT method.

Among the modern methods for surface treatment of metal surfaces is the electrospark alloying (ESA) method, which makes it possible to obtain surface structures having unique physico-mechanical and tribological properties at nano levels [11].

Compared with traditional surface modification technologies, the ESA method has several advantages: precise localisation, occurrence of a small thermal background and, as a result, absence of warping and deformation, a high degree of adhesion, simplicity and flexibility of technological process, environmental safety, etc. [12].

Despite the undeniable advantages, the ESA method has some disadvantages: relatively small and uneven thickness of the formed layer, its porosity and roughness, etc. [13].

The properties of the surface layers which have been obtained applying the contact and non-contact ESA methods in different environments, on various substrates and using electrodes made of various conductive materials are described in manuscript [14].

When the electrodes approach each other, the surfaces are subjected to the local actions of high pressures of the shock waves and temperatures [15]. In this case, the anode is rapidly heated, and a drop or a solid particle of the anode passes to the cathode. Fragments flying from the anode to the cathode are heated to a high temperature. The temperature of short-term heating of the surface micro volumes reaches (5–7)10^3^ °K. The spark discharge occurs in microscopically small volumes, and lasts from 50 to 400 microseconds. Holes and microbaths are formed on the cathode, wherein the materials of anode and cathode interact with each other and with the environment. The diffusion processes are activated here, which leads to the formation of new phases and to a change in the structures of the surface layers.

Broad opportunities for providing correctly directed changes in metal surface properties open up when graphite is used as an anode while performing the ESA process. According to [16], the choice of graphite as an electrode material is justified due to the number of its advantages. It is known that graphite in its free state is an excellent solid lubricant, as well as the fact that graphite, when in its bound state, namely, in the form of carbides, is represented by a hard wear-resistant phase, being quite resistant to many of aggressive media [17]. In some events, there is required simultaneous presence of the above properties.

When the ESA method is carried out using a graphite electrode, it is based on the process of diffusion, namely, on the process of saturation of the part surface with carbon. This is not accompanied by an increase in the size of the part; the surface is saturated with carbon due to diffusion, which fact gives a reason to compare it with a type of the CTT process, carburization (CESA) [18].

During the CESA process, strengthening a part surface occurs due to the diffusion-hardening process. The surface of the part is saturated with carbon at a sufficiently high temperature (up to 10,000 °C), followed by rapid cooling to room temperature.

In [19], the authors studied the CESA process in connection with steel surfaces. This process has a number of advantages when compared to the traditional methods carried out when using the CTT method. The main advantages of the CESA method are as follows: achieving 100% continuity for strengthening the surface, increasing the hardness of the surface layer of the part due to diffusion-hardening processes, the possibility of local processing (alloying can be carried out in certain places without protecting the rest of the surface of the part), etc. During the CESA, discharge energy in a range of 0.036 to 6.8 J and productivity of process in a range of 0.5 to 3.0 cm^2^/min are used.

Compared to carburizing and hardening processes, the nitriding process takes place at a lower temperature. A nitrided surface has a higher hardness, corrosion and wear resistance and also better polish ability. The properties of the nitrided surface practically remain unchanged upon reheating up to 500–600 °C, whereas, when the carburized and heat treated surface is heated to 250–290 °C, its hardness decreases [20].

In [21], to ensure the high durability of steel parts, a new, environmentally friendly and efficient process of nitriding the steel surfaces was developed. The process was implemented by the ESA method. The process includes the following steps: applying a special saturating technological medium, which is used as a paste-like nitrogenous component, onto the work surface and, without waiting for the above medium to dry, carrying out the treatment of the medium with an electrode-tool made of a material identical to the material of the steel part being processed.

When applying this process for nitriding steel C22 and steel C40, with an increase in the discharge energy from value 0.13 to 3.4 J, the following parameters are increased:the thickness of the “white” layer—from 10 μm to 40 μm and from 10 μm to 50 μm, respectively,its micro hardness—from 6228 to 8969 and from 6860 to 9160 MPa,the surface roughness (Ra)—from 0.9 μm to 6.2 µm and from 0.9 μm to 5.9 µm,and the coating continuity—from 50% to 70% and from 60% to 80%.

Replacing the substrate made of steel C22 with steel C40 does not introduce any significant changes into the quality of the surface layer. The wear resistance of the nitrided surfaces increases in comparison with specimens without treatment by 219% and 308%, respectively, for steel 20 and steel 40. The disadvantages of this process are the small thickness of the “white layer” and its relatively low micro hardness.

From [22], it is known that in order to obtain steel of high hardness, in the course of a nitriding process, as substrates, use is made of the aluminum-containing and improved steels 41CrAlMo7, 42CrMo4, 34CrAl6. This is confirmed in [23], according to which the steel being nitrided is distinguished by high endurance, wear resistance, corrosion resistance, hardness and other favorable properties. When being nitrided, the steel, which contains aluminum in its composition, forms strong nitrides, and the basis for producing nitrided steel of high hardness is steel of the following grades: 41CrAlMo7 and 34CrAl6. The disadvantages of this traditional method of nitriding are: the high cost of steel containing aluminum and the equipment used (minimally, the need for a nitriding plant), long process duration, the need to manufacture special technological equipment to protect the surfaces of the parts that are not subject to strengthening, etc.

Considering the advantages and disadvantages of the above-mentioned processes, a new process implementing the ESA method was proposed. The use of this process makes it possible to effectively carry out a nitriding process on steel part surfaces. The new process of nitriding, as well as the processes known from the prior art, are carried out with the use of the method of electrospark alloying and comprises such steps as:applying a saturating medium in the form of a paste-like nitrogenous component onto the surface being strengthened,without waiting for it to dry, conducting the electrospark alloying process with an electrode-tool made of a material identical to the material of the steel part being processed, but at the same time, in accordance with the new technical solution,before applying the saturating medium onto the surface to be strengthened, an aluminum layer is applied by the electrospark alloying (ESA) method at discharge energy of W_p_ = 0.13–6.8 J.

It should be noted that just those products which should correspond to the requirements of their surface layer features, such as high wear resistance and micro hardness, increased cyclic strength and corrosion resistance, are subject to the nitriding process.

A promising direction for increasing the resource of machine parts is the CTT process, namely, the nitrocarburizing process, which makes it possible to increase both mechanical properties (strength, hardness, fatigue strength) and corrosion resistance. Nitrocarburizing has a number of advantages over carburizing. These include: lower temperatures and a shorter process time are required for the saturation process; higher strength properties of the parts are obtained; in some cases, there is a possibility of replacing alloy steels with carbon steels; less deformation of the parts being processed is achieved. The nitrocarburized layer provides parts of sufficient viscosity and high strength. The nitrocarburized coatings have a high resistance to abrasion, which is associated with formation of nitrogen-containing phases [24]. Compared to the carburizing process, these advantages of nitrocarburizing provide a reduction in both electricity and natural gas consumption, an increase in the service life of the technological equipment and an increase in the durability and reliability of the machine parts.

The high-speed anodic nitrocarburizing process has certain advantages. These include: a short processing time; no need for special surface preparation; method precise localization [25].

In compliance with [26], the authors developed a process for strengthening the surfaces of the heat-treated steel parts, which comprises providing treatment with the use of the CESA method and differs in that nitrogen is supplied into the alloying zone. In this case, two processes, namely, the process by the CESA method and the process of nitriding, are conducted simultaneously, which, in fact, is the process of nitrocarburizing by the ESA method.

It should be noted that the implementation of this process requires the constant presence of nitrogen, and its rather large consumption significantly reduces the advantages thereof.

According to [27], a new environmentally safe and effective process of nitrocarburizing was proposed, which is carried out by the ESA method and comprises the step of applying a pasty carburizer which contains nitrogen-carbon components to the surface being strengthened, and that pasty carburizer, without waiting for it to dry, is treated with a graphite electrode-tool.

When nitrocarburizing the specimens of steel C22 and steel C40, with an increase in the discharge energy from 0.13 J to 3.4 J, the following parameters increase: the thickness of the “white layer”—from10 μm to 110 μm and of 20 μm to 120 μm, respectively; its micro hardness from 6.665 to 9.731 and from 7.135 to 9.932 MPa; the surface roughness (Ra) from 0.8 to 4.1 µm and from 0.9 to 4.7 µm; the coating continuity from 80 to 100 and from 90 to 100%. Replacing the substrate of steel C22 with steel C40 does not significantly affect surface layer quality.

At all researched discharge energies, the largest amounts of carbon and nitrogen are situated closer to the surface, and they decrease as the depths of their locations increase. Increase in the discharge energy of ESA from 0.13 to 3.4 J leads to an increase the parameters of the diffusion zone of carbon and nitrogen, for carbon from 20 to 45, for nitrogen from 20 to 55 μm.

The disadvantages of this process are: the small thickness of the “white layer” and its relatively low micro hardness.

According to [28], a new environmentally safe and efficient carburization process carried out by the ESA method has been proposed, which comprises the step of applying a pasty carburizer containing 80% graphite powder and 20% Vaseline to the surface being strengthened, while this pasty carburizer, without waiting for it to dry, is treated with a graphite electrode-tool.

It should be noted that as a result of analysis of the above works aimed at improving the quality of the surface layers of the steel parts, it was found that in terms of efficiency, environmental safety, cost, etc., the most promising technologies are those based on the ESA method. At the same time, because of using the step of applying a pasty carburizer which contains nitrogen-carbon components to the surface being strengthened, and due to the fact that these components show their maximum activity at different temperatures (500–900 °C), it is possible to carry out processing of steel products with the use of various types of treatments, namely, from virtually pure nitriding to nitrocarburizing and carburizing. In addition, ESA processing of the carbon steel surfaces being strengthened with the use of aluminum can improve the quality parameters of the surface layers.

Thus, the aim of this paper is to increase the operational performance of steel parts by developing a technology that provides control of the quality parameters of their surfaces using the range of the technologies from nitriding to nitrocarburizing and carburizing, thanks to the step of applying a pasty carburizer containing nitrogen-carbon components to the surface being strengthened and subsequent processing thereof by the ESA method.

## 2. Materials and Methods

Before applying the saturating media, an aluminum layer is applied onto the surface being strengthened with the use of the ESA method at a discharge energy of W_p_ = 0.13–6.80 J.

Samples were prepared from steels C22 and C40 of 15 × 15 × 8 mm size in a normalized and annealed state respectively.

For applying aluminum, the ESA unit of Elitron-52A model was used, which provides a discharge energy W_p_ in a range of 0.05 to 6.80 J. In this case, for the research, discharge energy values of W_p_ = 0.13; 0.52 and 3.40 J were used. Aluminum was applied with an electrode-tool in the form of an aluminum wire with a diameter of 3.0 mm, of AD, according to GOST 14838-78 (State Standard). For the ESA process with a material identical to the material of the steel part being processed, the same unit and the same modes of the discharge energy W_p_ = 0.13; 0.52; and 3.40 J were used.

In the course of the nitriding process, as a saturating medium, a pasty mixture was used, prepared by mixing with urea powder (~90%); in the course of nitrocarburizing, the pasty mixture was made of urea (45%), yellow blood salt (45%), and Vaseline (10%), and in the course of the CESA process, the pasty mixture was made of graphite powder (80%) and Vaseline (20%).

The optical microscope Neofot-2 was used for metallographic studies of the prepared specimens. It was possible to determine the quality of the layer, its continuity, thickness and structure of zones, namely, “white” layer and the diffusion zone.

The durametric analysis of the micro hardness distribution in the surface layer and over the depth of the section beginning from the surface was also performed. The micro hardness was measured on the PMT-3 micro hardness tester by indenting a diamond pyramid under a load of 0.05 N, according to GOST 9450-76.

At all stages of processing, the surface roughness was measured using the profilograph-profilometer of 201 model of the “Caliber” plant production. The results were recorded using a special attachment.

A saturating medium in the form of a paste was applied to the surfaces of the specimens after processing the surfaces by the ESA method with the use of an electrode made of aluminum wire of 3 mm diameter, of AD mark, according to GOST 9450-76. Then, without waiting for the above mentioned surface to dry, using the ESA method, processing was carried out by an electrode-tool in the form of a 3 mm diameter wire made of a material identical to the material of the steel part being processed during nitriding or by an electrode-tool in the form of a graphite rod of EG-4 mark with the size of 3 × 3 × 25 mm during nitrocarburizing and the CESA processes.

X-ray diffraction studies were carried out on a DRON-3 diffractometer in CoK*α*-radiation. The voltage and anode currents were 30 kV and 24 mA, respectively. X-ray diffraction patterns were taken by step-by-step scanning with an exposure for 2 s at every point. The experimental results were processed using Powdercell 2.4 software for the Rietveld refinement analysis of X-ray diffraction patterns of the mixture of polycrystalline phase components.

## 3. Results and Discussion

Figure 1 and Figure 2 shows the microstructures of the surface layers of the specimens made of steel C22. Figure 1 shows the microstructure after carrying out the ESA process with the use of an electrode-tool made of aluminum at the discharge energy of W_p_ = 3.4 J, and Figure 2 shows the microstructure after nitriding with the use of the ESA process. This comprised the step of forming the aluminum sublayer by alloying with an aluminum electrode-tool at W_p_ = 3.40 J, and also that of applying the nitrogen-containing paste onto the surface to be processed, and the step of subsequent alloying with an electrode-tool made of steel C22 at W_p_ = 0.13; 0.52 and 3.40 J, respectively. In both cases (Figure 1 and Figure 2), the coatings each consisted of three areas, namely, the “white layer”, the diffusion zone, and the base metal.

In the scientific literature, “white layers” are characterized by many scientists as martensitic-austenitic, with a fine-needled structure, having a high density of defects in the crystal structure. The formation of fine-needled structures in the “white layer” is caused, as a rule, by the substructural interaction of defects in the crystal structure. The diffusion zone has areas of incomplete recrystallization; along with grains formed as a result of recrystallization, grains of the original metal are present. The base metal (steel C22) has a ferrite-pearlite structure.

Figure 3 shows the graph of the surface layer micro hardness distribution. On the graph: 0—after the ESA process by aluminum electrode-tool at W_p_ = 3.4 J; 1, 2, and 3—after nitriding by the ESA method (with aluminum sublayer) at W_p_ = 3.4 J, respectively, with the ESA process by electrode-tool made of steel C22, using discharge energy W_p_ = 0.13; 0.52 and 3.40 J. On the surface, the maximum hardness is about 9700 MPa, and further from the surface it decreases. Such a change in microhardness is due to the diffusion of electrode material into the substrate, changing the structural phase composition of the layer.

Table 1 summarizes the results of measuring the thickness, micro hardness and continuity of the “white” layer, as well as the surface roughness of the specimen of steel C22 after nitriding, without and with the aluminum sublayer, which was obtained with the use of the ESA process at W_p_ = 0.13; 0.52 and 3.4, followed by the process of applying the nitrogen-containing paste and performing the ESA process by the electrode-tool made of steel C22 at the discharge energy of W_p_ = 0.13; 0.52 and 3.4 J. In addition, the Table presents the results of measuring the thickness, micro hardness and continuity of the “white” layer, as well as the surface roughness of the specimen of steel C40 after the ESA process by the electrode-tool made of aluminum at W_p_ = 3.40 J and the ESA process by the electrode-tool made of steel C40 at the discharge energy of W_p_ = 0.13; 0.52 and 3.40 J.

It should be noted that the coating parameters depend on the ESA discharge energy. With an increase in W_p_, the continuity of the “white” layer, its thickness and microhardness increased. However, as expected, the surface roughness increased in this case.

Figure 4 shows the microstructure of the steel C40 specimen surface layer after the nitrocarburizing process by the ESA method, comprising the steps of: obtaining the aluminum sublayer by the ESA process with the use of the aluminum electrode-tool at the discharge energy of W_p_ = 3.40 J; applying a nitrogen-containing paste to the surface being processed; subsequent alloying the processed surface by a graphite electrode-tool at W_p_ = 0.13; 0.52 and 3.40 J, respectively. Compared to the process of nitriding, after the process of nitrocarburizing, the thicknesses of the “white layer” and the diffusion zone are larger. Apparently, this is due to the fact that nitrocarburizing is carried out with a graphite electrode-tool, unlike nitriding, which uses a steel electrode-tool. A section of incomplete recrystallization is visible on the microstructure. For C40 steel, this is heating to a temperature of 730–755 °C. The metal in this area undergoes incomplete recrystallization. The base metal (steel C40) has a ferrite-pearlite structure.

Figure 5 shows the graph of the micro hardness distribution in the nitrocarburized layer. On the graph: 1, 2 and 3—with the sublayer of aluminum obtained with the use of the ESA method by the aluminum electrode-tool at W_p_ = 3.40 J and also with the use of the ESA method by the graphite electrode-tool at the discharge energy of W_p_ = 0.13; 0.52; and 3.4 J, respectively.

The results of measuring the thickness, micro hardness and continuity of the “white layer”, as well as the surface roughness of the specimen of steels C22 and C40 after nitrocarburizing are summarized in Table 2. After the nitrocarburizing process the thickness parameter increases the more intensely, the higher the discharge energy at which the aluminum sublayer is applied. The microhardness and continuity of the “white layer” increased with preliminary ESA aluminum, but the roughness changed insignificantly.

Table 3 summarizes the results of measuring the thickness, micro hardness and continuity of the “white layer”, as well as the magnitude of the surface roughness of the specimen of steel C22 after carburizing by the ESA method, which consisted of alloying the surface by the ESA method with the aluminum electrode-tool at W_p_ = 0.13; 0.52; and 3.40 J, applying a carbon-containing paste to the surface being processed, and processing the obtained surface with the use of the ESA method by a graphite electrode-tool at the discharge energy of W_p_ = 0.13; 0.52 and 3.4 J. The thickness, microhardness of the “white layer” increased with preliminary ESA aluminum and continuity did not change. After the carburization process, the thickness parameter increased with increasing discharge energy at which the aluminum sublayer was applied.

The studies of the phase composition of the obtained coatings on steel C40 have shown that, at the process of nitriding, the phase composition is represented by a bcc solid solution, obviously alloyed with ferrite, and cubic iron nitride (Figure 6a). Most probably, aluminum dissolves in ferrite and nitride. In addition, there is a possibility of the formation of the sublayer made of the aluminum-containing phases between the substrate and the nitrogen-containing coating. This is evidenced by the results of measuring the micro hardness (Figure 3, dependence 2), at alloying, the micro hardness at the distance of ~70 μm from the surface increases, as compared with the values on the surface.

After nitrocarburizing by the ESA method, the presence of a bcc solid solution, cubic iron nitride, and iron carbide (Cohenite) was revealed (Figure 6b). Clearly, in the represented phases, carbon atoms can be partially replaced by nitrogen atoms. In addition, it is likely that aluminum dissolves in ferrite and iron nitride, and also it is located in the sublayer between the coating and the substrate.

After carburizing the specimens with the aluminum sublayers obtained after the ESA process, X-ray diffraction analysis revealed the presence of a bcc solid solution, apparently alloyed ferrite, the diffraction maxima of which were completely overlapped by the tetragonal iron carbide C_0.055_Fe_1.945_. The rest of the diffraction maxima are related to the rhombohedral aluminum carbide Al_4_C_3_.

The analysis of the phase composition of the coatings indicates that the presence of the aluminum sublayer ensures the formation of the phases, the availability of which results in a significant increase in hardness, and, in addition, provides for increasing the thickness and quality of the surface layers.

## 4. Conclusions

Based on the analyses described in this manuscript, the authors present their conclusions as follows:For steel C22 and steel C40, use of nitriding, nitrocarburizing and CESA processes, with an increase in the discharge energy (W_p_), increases the thickness, micro hardness and continuity of the “white layer”, as well as the magnitude of the surface roughness.In the course of nitriding, nitrocarburizing, and CESA processing steel C22 and steel C40, preliminary processing with the use of the ESA method with aluminum increases the thickness, micro hardness and continuity of the “white layer”, while the roughness changes insignificantly.Preliminary processing with the use of the ESA method with aluminum at W_p_ < 0.13 J, does not result in noticeable changes in the qualitative parameters of the surface layer being formed at all types of processing, and at W_p_ > 3.40 J, it is accompanied by a sharp increase in the roughness of the surface layer. For practical application, the recommended discharge energy parameters are in the range of W_p_ = 0.13–3.40 J for nitriding and nitrocarburizing, and in the range of W_p_ = 0.13–4.60 J for the CESA process.The analysis of the phase composition indicates that the presence of the aluminum sublayer leads to the formation of the aluminum-containing phases, resulting in a significant increase in the hardness, and, in addition, an increase in the thickness and quality of the surface layers.The proposed methods can be used to strengthen the surface layers of the critical parts and their elements for compressor and pumping equipment: the outer and inner surfaces of the protective bushings for the floating seals, the end surfaces of their rings, and the mating parts of the cases and covers; the bearing necks of the centrifugal machine rotor shafts; the piston rods, etc.

## Figures and Tables

**Figure 1 materials-15-06085-f001:**
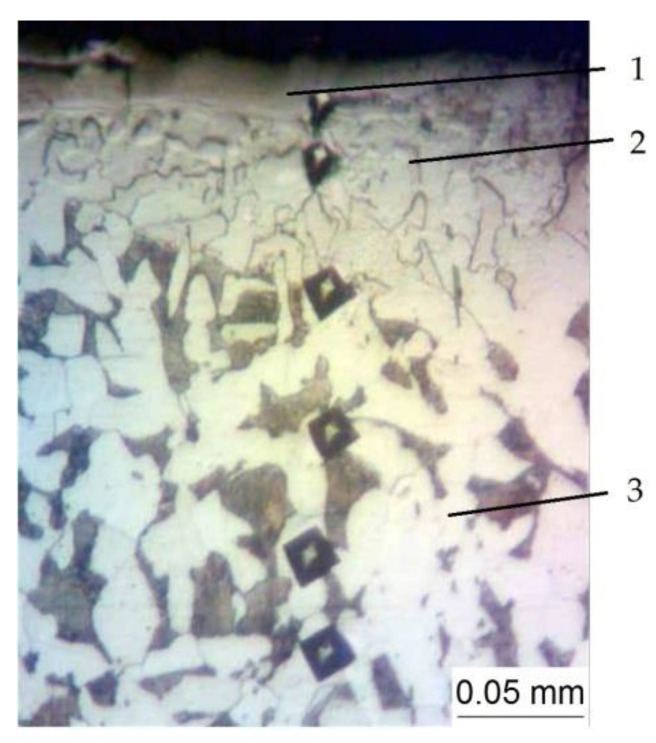
The microstructure of the steel C22 specimen surface layer after the ESA process with the use of the aluminum electrode-tool: 1—”white layer”, 2—the diffusion zone, 3—the base metal.

**Figure 2 materials-15-06085-f002:**
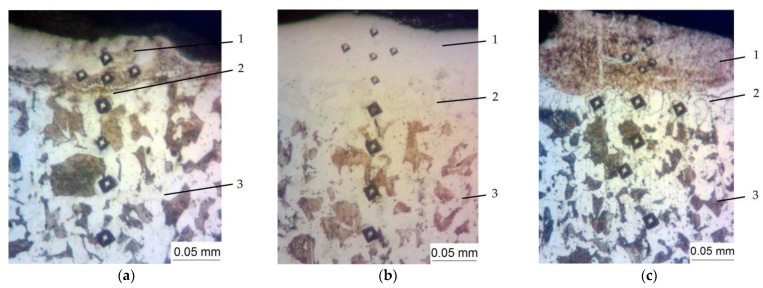
The microstructures of the nitrided surface layers of the steel C22 specimens having the aluminum sublayer and processed with the use of the ESA method by the electrode-tool made of steel C22 at the discharge energy (W_p_): (**a**)—W_p_ = 0.13; (**b**)—W_p_ = 0.52 and (**c**)—W_p_ = 3.40 J, 1—“white layer”, 2—the diffusion zone, 3—the base metal.

**Figure 3 materials-15-06085-f003:**
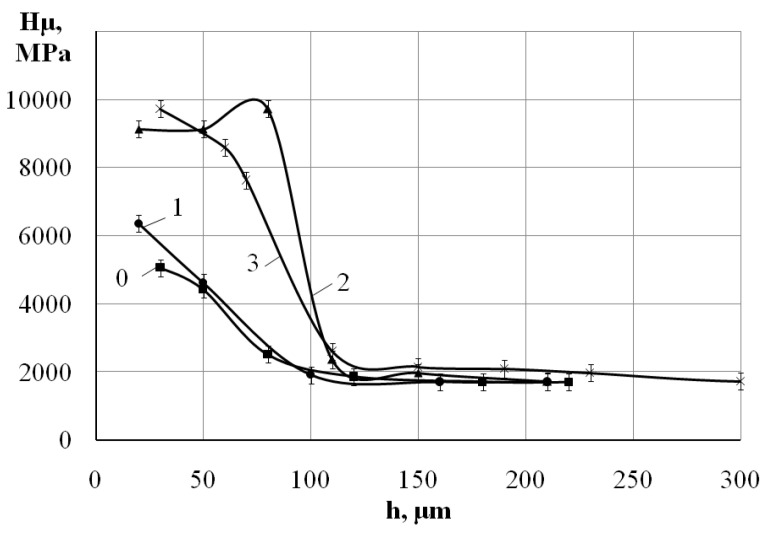
The distribution of the micro hardness over the depth of the layer, beginning from the surface of the specimen made of steel C22: 0—the ESA process by aluminum at W_p_ = 3.40 J; after nitriding process with the use of the nitrogen-containing paste and alloying by the electrode-tool made of steel C22 at the discharge energy (W_p_): ***1***—W_p_ = 0.13; ***2***—W_p_ = 0.52, and ***3***—W_p_ = 3.4 J.

**Figure 4 materials-15-06085-f004:**
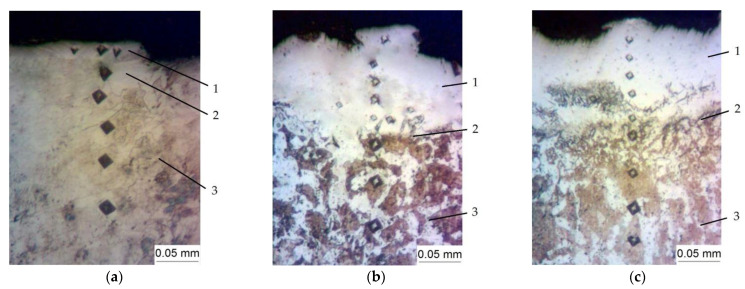
The microstructures of the nitrocarburized surface layers of the specimens made of steel C40 with the aluminum sublayers and processed with the use of the ESA method by the electrode-tool of steel C40 at the discharge energy (W_p_): (**a**)—W_p_ = 0.13; (**b**)—W_p_ = 0.52 and (**c**)—W_p_ = 3.40 J, 1—”white layer”, 2—the diffusion zone, 3—the base metal.

**Figure 5 materials-15-06085-f005:**
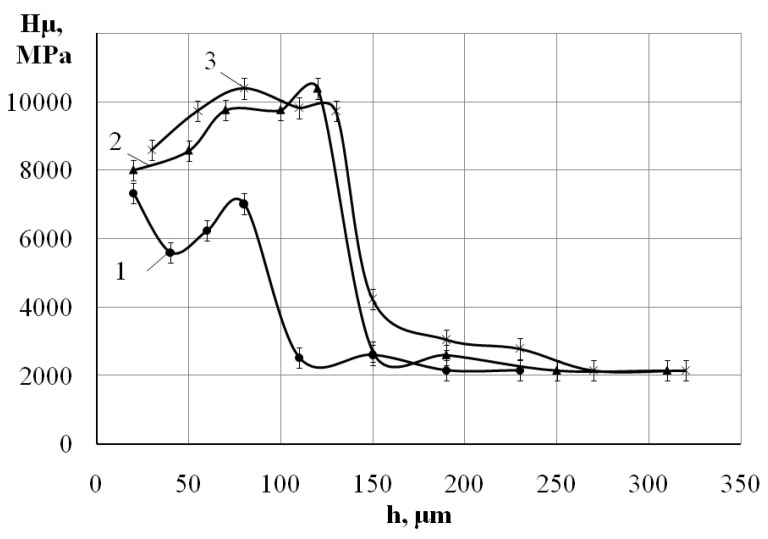
The distribution of the micro hardness over the depth of the nitrocarburized layer beginning from the surface of the steel C40 specimen after the ESA process by aluminum electrode-tool at W_p_ = 3.4 J, applying a nitrogen-containing paste to the surface being processed, and alloying the obtained surface by the graphite electrode-tool at the discharge energy (W_p_): ***1—***W_p_ = 0.13; ***2—***W_p_ = 0.52; and ***3—***W_p_ = 3.40 J.

**Figure 6 materials-15-06085-f006:**
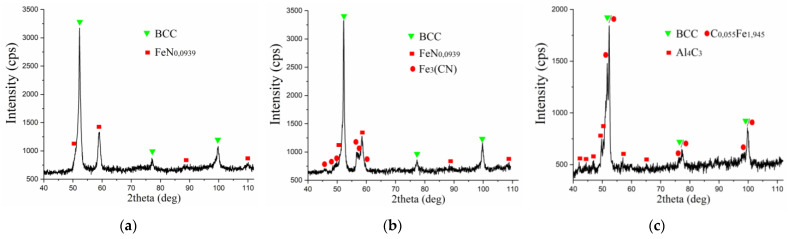
The X-ray diffraction patterns of the surface layers of the specimens made of steel C40 with the aluminum sublayer after processing with the use of the ESA method at W_p_ = 0.52 J: (**a**)—after nitriding, (**b**)—after nitrocarburizing, (**c**)—after carburizing.

**Table 1 materials-15-06085-t001:** The qualitative parameters of the nitrided layers obtained by the ESA process on steel C22 and steel C40.

Discharge Energy, J	Thickness of the “White” Layer, µm	Microhardness of the “White” Layer, MPa	Roughness, μm			Continuity of the “White” Layer, %
			Ra	Rz	Rmax	
Without aluminum sublayer						
Steel C22						
0.13	10–20	6228	0.9	2.1	7.5	50
0.52	10–20	7150	1.3	2.3	9.3	60
3.40	30–40	8969	6.2	16.3	40.6	70
Steel C40						
0.13	10–25	6860	0.9	2.0	7.1	60
0.52	10–30	7450	1.4	2.2	8.3	70
3.40	30–50	9160	5.9	17.3	34.6	80
With aluminum sublayer at W_p_ = 0.13 J						
Steel C22						
0.13	10–25	6300	0.8	2.1	7.3	50
0.52	10–30	7250	1.6	2.3	9.1	65
3.40	30–40	9100	5.8	15.3	38.2	75
With aluminum sublayer at W_p_ = 0.52 J						
Steel C22						
0.13	20–35	6320	0.9	2.2	7.3	55
0.52	25–50	7450	1.6	2.2	9.2	70
3.40	35–70	9300	5.9	15.7	38.6	80
With aluminum sublayer at W_p_ = 3.4 J						
Steel C22						
0.13	60–70	6350	1.1	2.3	7.4	70
0.52	60–80	9721	1.5	2.2	9.5	80
3.40	90–110	9721	6.5	16.1	39.1	90
With aluminum sublayer at W_p_ = 0.13 J						
Steel C40						
0.13	60–80	7050	0.9	2.1	7.3	80
0.52	60–90	9850	1.3	2.4	8.4	85
3.40	100–130	9910	5.9	17.3	35.7	90

**Table 2 materials-15-06085-t002:** The qualitative parameters of the nitrocarburized (N + C) layers obtained by the ESA method on steel C22 and steel C40.

Discharge Energy, J	Thickness of the “White” Layer, µm	Microhardness of the “White” Layer, MPa	Roughness, μm			Continuity of the “White” Layer, %
			Ra	Rz	Rmax	
Without aluminum sublayer						
Steel C22						
0.13	10–20	6665	0.9	1.9	6.5	80
0.52	30–40	7689	1.2	2.1	8.1	90
3.40	80–110	9731	4.1	11.3	25.1	100
Steel C40						
0.13	20–30	7135	0.9	2.2	7.3	80
0.52	30–50	7920	1.3	2.7	8.7	90
3.40	80–120	9930	4.7	16.2	35.1	100
With aluminum sublayer at W_p_ = 0.13 J						
Steel C22						
0.13	10–25	6850	0.9	1.9	7.5	80
0.52	10–30	8010	1.5	2.1	8.3	90
3.40	30–40	9930	5.6	12.4	28.5	100
With aluminum sublayer at W_p_ = 0.52 J						
Steel C22						
0.13	20–35	6900	0.9	2.0	7.5	85
0.52	30–45	8300	1.7	2.2	8.7	100
3.40	50–70	9800	5.7	12.7	28.7	100
With aluminum sublayer at W_p_ = 3.4 J						
Steel C22						
0.13	60–70	7150	1.1	2.1	7.5	90
0.52	60–80	9721	1.6	2.3	9.1	100
3.40	90–110	10,050	5.9	13.3	28.9	100
With aluminum sublayer at W_p_ = 0.13 J						
Steel C40						
0.13	50–70	7320	1.2	2.2	7.3	100
0.52	60–80	10,380	1.9	2.9	8.7	100
3.40	80–110	10,380	6.3	16.8	31.1	100

**Table 3 materials-15-06085-t003:** The qualitative CESA parameters obtained on steel C22 and steel C40.

Discharge Energy, J	Thickness of the “White” Layer, µm	Microhardness of the “White” Layer, MPa	Roughness, μm Ra	Continuity of the “White” Layer, %
Without aluminum sublayer				
Steel C22				
0.9	50–70	9932	1.0	100
2.6	80–100	10,796	3.7	
4.6	100–230	10,796	4.8	
Steel C40				
0.9	60–80	11,351	1.0	100
2.6	90–110	11,787	3.8	
4.6	130–240	11,824	4.7	
With aluminum sublayer at W_p_ = 0.13 J				
Steel C22				
0.9	55–75	10,153	1.0	100
2.6	85–110	11,930	3.6	
4.6	100–230	11,995	4.8	
With aluminum sublayer at W_p_ = 0.52 J				
Steel C22				
0.9	60–80	10,320	1.0	100
2.6	90–120	11,950	3.8	
4.6	130–240	12,100	4.8	
With aluminum sublayer at W_p_ = 3.4 J				
Steel C22				
0.9	70–90	10,370	1,0	100
2.6	100–130	12,050	3.7	
4.6	140–260	12,200	4.9	
With aluminum sublayer at W_p_ = 0.13 J				
Steel C40				
0.9	80–100	11,760	1.0	100
2.6	110–140	12,240	3.7	
4.6	140–280	12,375	4.9	

## Data Availability

The data presented in this study are available on request from the corresponding author. The data are not publicly available due to no technical possibilities.

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
