# Peer review of "Technological Features for Controlling Steel Part Quality Parameters by the Method of Electrospark Alloying Using Carburezer Containing Nitrogen—Carbon Components"

_materials, 2022, doi:10.3390/ma15176085_

Round 1
Reviewer 1 Report
The manuscript presents an interesting study about the use of the electrospark alloying method using carburezer containing nitrogen-carbon components for producing coatings. The paper needs major revisions before it is processed further, some comments follow:
Abstract:
The abstract is too long, please make it shorter.
Introduction:
The introduction section must be improved. In the introduction section, a comprehensive and exhaustive review of the state of the art in the field of the study must be provided. Please introduce and discuss more previous works, and highlight the experiments and results published previously.
Also, multiple citations have been introduced in bulk form "[3-8]", "[9-13]", "[14-16]" etc. and not distributed in the text in accordance with the affirmations that must be supported. Please introduce citations at a specific position to assure a clear correspondence between the affirmations from the introduction section and the previous publication. Moreover, to avoid this type of citing, please cite review type of studies.
In the last paragraph please highlight the novelty and importance of this study.
Materials and methods
Almost all the information written in this section is related to the state of art and must be moved to the introduction section.
This section must contain just information about the methodology (methods, parameters etc.). Also, must contain information about the chemical composition of the substrate, the difference between samples etc. Please modify all the materials and methods section.
Results and discussion
Figure 2. Introduce figure labels to highlight the zone of interest for the reader. Also, introduced in the figure title what a, b, and c represent. Also, these comments are also for figure 4.
Figure 6. Introduced in the figure title what a, b, and c represent
The discussions are poor. Please improve.
References
There are too many self-citations please remove the unnecessary ones.
Author Response
- «Abstract: The abstract is too long, please make it shorter.
Agree with Remark. We made the necessary additions.
- «Introduction: The introduction section must be improved. In the introduction section, a comprehensive and exhaustive review of the state of the art in the field of the study must be provided. Please introduce and discuss more previous works, and highlight the experiments and results published previously.
Also, multiple citations have been introduced in bulk form "[3-8]", "[9-13]", "[14-16]" etc. and not distributed in the text in accordance with the affirmations that must be supported. Please introduce citations at a specific position to assure a clear correspondence between the affirmations from the introduction section and the previous publication. Moreover, to avoid this type of citing, please cite review type of studies.
In the last paragraph please highlight the novelty and importance of this study.»
Agree with Remark. We made the necessary additions.
- «Materials and methods. Almost all the information written in this section is related to the state of art and must be moved to the introduction section.
This section must contain just information about the methodology (methods, parameters etc.). Also, must contain information about the chemical composition of the substrate, the difference between samples etc. Please modify all the materials and methods section.»
Agree with Remark. We made the necessary additions.
- «Results and discussion. Figure 2. Introduce figure labels to highlight the zone of interest for the reader. Also, introduced in the figure title what a, b, and c represent. Also, these comments are also for figure 4.
Figure 6. Introduced in the figure title what a, b, and c represent.
The discussions are poor. Please improve. »
Agree with Remark. We made the necessary additions.
- «References. There are too many self-citations please remove the unnecessary ones.»
Agree with Remark. We made the necessary additions.
Thank you for your comments
Reviewer 2 Report
The format of the article is in a mess. Many sentences are in a continuous format without blank between words.
The introduction is not enough to support the importance of the present work.
The experiment part is a bit like the introduction part.
I recommend a thoroughly format revvision and resubmission.
Author Response
The format of the article is in a mess. Many sentences are in a continuous format without blank between words.
The introduction is not enough to support the importance of the present work.
The experiment part is a bit like the introduction part.
I recommend a thoroughly format revvision and resubmission.
Agree with Remark. We made the necessary additions.
Thank you for your comments
Reviewer 3 Report
I have read the article entitled “Technological Features for Controlling Steel Part Quality Parameters by the Method of Electro Spark Alloying (ESA) Using Carburizer Containing Nitrogen - Carbon Components” which is having some scientific contributions by the authors. Authors have tried to improve the performance of coating the steel parts (C22 and C40) in carburizing and nitriding of conventional methods by adopting electro spark alloying approach. However, the present form of article is not well written, not in well-structured and having lot of grammatical mistakes. I have observed the following comments which the authors have to address it before it is being considered for its publications.
1. In the abstract, the authors have mentioned that the performance (hardness) of steel products are increased after CESA approach. What is the reason behind it which is missing in the abstract?
2. Moreover, the present form of abstract is too lengthy one which has to be shortened based on their specific work, outcomes and findings
3. The introduction part is too short. Some more recent literatures (last 7 years) related to their work are to be incorporated
4. The main objectives of the present work are to be added at the end of introduction section
5. Have the authors considered the cost and energy calculations for adopting this ESA approach in addition to the regular carburizing and nitriding approaches? Need to be addressed
6. The language of the entire manuscript is not up to the level of standard. It has to be checked by native English speaker. For instance: “Depending on operating conditions and part requirements, one of the two methods is used”. This sentence is having lot of grammatical mistakes. In similar manner, there are several sentences having grammatical mistakes.
7. Materials and methods section is too long. 80% of the part is to be moved to introduction section. Here, the authors are asked to write only on their experimental part. All other stories are to be removed to introduction part or removed completely.
8. Various features / phases observed from optical microstructures of Figures 1 and 2 are to be incorporated.
9. Cross-sectional images of micro/macro structures showing the substrate and coating zone are to be incorporated
10. Coating zone and substrate zone are to be highlighted/marked in Figure 3.
11. In Figure 3, there are several lines with different symbols showing the hardness profiles. However, the plot is not mention the sample condition. It has to be checked.
12. Various features / phases observed from optical microstructures of Figure 4is to be incorporated
Author Response
- In the abstract, the authors have mentioned that the performance (hardness) of steel products are increased after CESA approach. What is the reason behind it which is missing in the abstract?
Agree with Remark. We made the necessary additions.
- Moreover, the present form of abstract is too lengthy one which has to be shortened based on their specific work, outcomes and findings
Agree with Remark. We made the necessary additions.
- The introduction part is too short. Some more recent literatures (last 7 years) related to their work are to be incorporated
Agree with Remark. We made the necessary additions.
- The main objectives of the present work are to be added at the end of introduction section
Agree with Remark. We made the necessary additions.
- Have the authors considered the cost and energy calculations for adopting this ESA approach in addition to the regular carburizing and nitriding approaches? Need to be addressed
Consideration of the cost and energy calculations of ESA method is not the aim of this article. It is known that the energy consumption of the "Elitron-52A" Installation is 0.3-2 kW. Shaft furnace СШЦМ-8.26/10 for carburizing, nitriding and carbonitration of steel consumes 90±5 kW (https://bortek.ub.ua/ru/goods/view/11880310/all/shahtnaya-termicheskaya-pech-dlya-cementacii-stali-sshcm-8-2610-kupit-v-ukraine/).
Energy efficiency of ESA method is not in dispute. Boris Lazorenko, the founder of the ESA method, confirmed its energy-saving nature.
We have introduced into production ESA methods as energy- and resource-saving, and an economic effect has been obtained.
- The language of the entire manuscript is not up to the level of standard. It has to be checked by native English speaker. For instance: “Depending on operating conditions and part requirements, one of the two methods is used”. This sentence is having lot of grammatical mistakes. In similar manner, there are several sentences having grammatical mistakes.
Agree with Remark. We made the necessary additions.
- Materials and methods section is too long. 80% of the part is to be moved to introduction section. Here, the authors are asked to write only on their experimental part. All other stories are to be removed to introduction part or removed completely.
Agree with Remark. We made the necessary additions.
- Various features / phases observed from optical microstructures of Figures 1 and 2 are to be incorporated.
Agree with Remark. We made the necessary additions.
- Cross-sectional images of micro/macro structures showing the substrate and coating zone are to be incorporated
Agree with Remark. Introduced figure labels to highlight the zones.
- Coating zone and substrate zone are to be highlighted/marked in Figure 3.
Agree with Remark. Introduced figure labels to highlight the zones.
- In Figure 3, there are several lines with different symbols showing the hardness profiles. However, the plot is not mention the sample condition. It has to be checked.
C22 steel samples were used in the normalized state of hardness HB165-180. We made the necessary additions in chapter “Materials and Methods”.
- Various features / phases observed from optical microstructures of Figure 4is to be incorporated
Agree with Remark. We made the necessary additions.
Thank You for Your time.
Round 2
Reviewer 1 Report
The authors didn't address all of my comments and did not provide detailed replace as the journal policy requires.
Author Response
The authors tried to take into account all the comments of the Reviewer regarding the text. However, some elements may have remained unchanged due to recommendations from two other Reviewers. However, all editorial comments will be corrected in accordance with the recommendations and policy of the journal at the stage of accepting the article for publication.
Reviewer 2 Report
(1)Add data supported progress in the abstract, not merely "XXX increases";
(2)The "Hμ" stands for what kind of hardness? 2000 is normal for C 22 ?
Author Response
- Add data supported progress in the abstract, not merely "XXX increases".
There is too much data. We add information in text: see Table 1-3
- The "Hμ" stands for what kind of hardness? 2000 is normal for C 22 ?
C22 steel samples were used in the normalized state of hardness HB165-180 (Hμ =1690-1710 MPa). Perhaps the scale in the figure 3 is not well chosen.
Reviewer 3 Report
The authors have revised the manuscript based on my previous comments and hence, I am recommending to accept this revised version
Author Response
Thank You for Your review.
Round 3
Reviewer 1 Report
The authors addressed almost all my comments. The paper can be accepted in the present form.